# Diphlorethohydroxycarmalol (DPHC) Isolated from the Brown Alga *Ishige okamurae* Acts on Inflammatory Myopathy as an Inhibitory Agent of TNF-α

**DOI:** 10.3390/md18110529

**Published:** 2020-10-26

**Authors:** Seo-Young Kim, Ginnae Ahn, Hyun-Soo Kim, Jun-Geon Je, Kil-Nam Kim, You-Jin Jeon

**Affiliations:** 1Department of Marine Life Science, Jeju National University, Jeju 63243, Korea; kimsy11@kbsi.re.kr (S.-Y.K.); gustn783@mabik.re.kr (H.-S.K.); wpwnsrjs@naver.com (J.-G.J.); 2Chuncheon Center, Korea Basic Science Institute (KBSI), Chuncheon 24341, Korea; knkim@kbsi.re.kr; 3Department of Food Technology and Nutrition, Chonnam National University, Yeosu 59626, Korea; gnahn@chonnam.ac.kr; 4Department of Applied Research, National Marine Biodiversity Institute of Korea, 75, Jangsan-ro 101-gil, Janghang-eup, Seocheon 33662, Korea

**Keywords:** inflammation, pro-inflammatory cytokines, myopathy, inflammatory myopathy, marine algae, phlorotannin

## Abstract

Inflammation affects various organs of the human body, including skeletal muscle. Phlorotannins are natural biologically active substances found in marine brown algae and exhibit anti-inflammatory activities. In this study, we focused on the effects of phlorotannins on anti-inflammatory activity and skeletal muscle cell proliferation activity to identify the protective effects on the inflammatory myopathy. First, the five species of marine brown algal extracts dramatically inhibited nitric oxide (NO) production in lipopolysaccharide (LPS)-induced RAW 264.7 cells without toxicity at all the concentrations tested. Moreover, the extracts collected from *Ishige okamurae* (*I*. *okamurae*) significantly increased cell proliferation of C2C12 myoblasts compared to the non-treated cells with non-toxicity. In addition, as a result of finding a potential tumor necrosis factor (TNF)-α inhibitor that regulates the signaling pathway of muscle degradation in *I*. *okamurae*-derived natural bioactive compounds, Diphlorethohydroxycarmalol (DPHC) is favorably docked to the TNF-α with the lowest binding energy and docking interaction energy value. Moreover, DPHC down-regulated the mRNA expression level of pro-inflammatory cytokines and suppressed the muscle RING-finger protein (MuRF)-1 and Muscle Atrophy F-box (MAFbx)/Atrgoin-1, which are the key protein muscle atrophy via nuclear factor-κB (NF-κB), and mitogen-activated protein kinase (MAPKs) signaling pathways in TNF-α-stimulated C2C12 myotubes. Therefore, it is expected that DPHC isolated from IO would be developed as a TNF-α inhibitor against inflammatory myopathy.

## 1. Introduction

Cytokines are intracellular signaling molecules, potent mediators of a number of cell functions and are essential in coordinating inflammatory responses [1]. In skeletal muscle, tumor necrosis factor (TNF)-α, one of the pro-inflammatory cytokines, influences satellite cell proliferation and accelerates the G1 to S phase transition [2]. Moreover, in dystrophic muscle, elevated levels of TNF-α inhibit the regenerative potential of satellite cells and are associated with loss of muscle [3,4]. Excessive TNF-α release enhances protein degradation of insulin-like growth factor I (IGF-1) and induces early activation of the atrogin-1 gene expression [5]. The loss of muscle by inflammation is called inflammatory myopathy, muscle wasting, and muscle atrophy. Therefore, TNF-α inhibitors are developed and used as anti-inflammatory and therapeutic agents. However, it is specific to the treatment of rheumatism, dry arthritis, and spondylitis. Moreover, it has the characteristics of injection, and the injection site is known to have side effects such as fever, pain, itching, and virus infection [6].

Marine organisms are rich sources of structurally diverse bioactive compounds with various biological activities. Especially, marine brown algae contain a variety of bioactive substances including phlorotannins, polysaccharide, and pigments and exhibit different biological activities [7,8] and potential health benefits such as antioxidant activity [9], anti-inflammatory activity [10], anti-cancer activity [11], antidiabetic activity [12], antihypertensive effects [13], and anti-obesity activity [14]. Especially, phlorotannins found only in brown algae, is a polyphenolic compound, the most important group of biological substances that determine the value of functional foods or pharmaceuticals [15]. These phlorotannins are among the most common classes of secondary metabolites derived from a unit of polymerized phloroglucinol [15]. However, studies on its effect of muscle growth and myopathy are scarce. Kunkel et al. [16] reported that ursolic acid increases skeletal muscle, and Kim et al. [17] revealed that carnosic acid induces muscle growth and differentiation through an in vivo study. Ursolic acid carnosic acid are natural polyphenols obtained from apple peel, and rosemary, respectively. From these studies, we considered that polyphenols contained in marine algae would also be positive for muscle growth, so we isolated polyphenol from marine algae and searched that act as a potential TNF-α inhibitor in muscle from them. Therefore, in this study, a prospective TNF-α inhibitor among diverse natural compounds from marine algae was searched.

## 2. Results

### 2.1. Effects of Marine Brown Algal Methanol Extracts on Nitric Oxide (NO) Production against Lipopolysaccharides (LPS)-Stimulated RAW 264.7 Cells

Prior to the evaluation of the NO inhibitory activity of the methanol extracts from marine brown algae, their cytotoxic effects on the viability of RAW 264.7 cells were examined. All the methanol extracts showed no cytotoxic effect on RAW 264.7 cells at the tested concentrations (6.25, 12.5, 25, 50, 100, and 200 µg/mL, data not shown). Thus, those concentrations were used in subsequent experiments. To examine the potential anti-inflammatory properties of the five marine brown algal methanol extracts on LPS-induced NO production in RAW 264.7 cells, cells were treated with or without extracts (6.25, 12.5, 25, 50, 100, and 200 µg/mL) for 24 h. NO production in the culture supernatants was measured by the Griess assay. The level of NO production is significantly increased in the LPS-treated cells compared to the untreated cells. However, all the methanol extracts significantly inhibited the LPS-induced NO production (Figure 1). All methanolic extracts did not affect the cytotoxic of RAW 264.7 cells at the tested concentrations except for *Myelophcus caespitosus* extract (MCE) (Figure 2).

### 2.2. Skeletal Muscle Cell Proliferation Activities of Marine Brown Algalmethanol Extracts on C2C12 Cells

Prior to evaluating the cell proliferation effect of marine brown algal methanol extracts, their cytotoxic effects on the viability of C2C12 myoblasts were examined with 3-(4-5-dimethyl-2yl)-2-5-diphynyltetrasolium bromide (MTT) assay and all the extracts showed a non-cytotoxic effect on C2C12 cells at the concentrations tested (Figure 3). Thus, those concentrations were used in subsequent experiments. The skeletal muscle cell proliferation activity of the different marine brown algal methanol extracts was measured using 5-bromo-2’-deoxyuridine (BrdU) cell proliferation assay. BrdU cell proliferation assay is a non-isotopic assay for the in vitro quantitative detection of newly synthesized DNA of actively proliferating cells. Among the extracts, *Ishige okamurae* (*I*. *okamurae*) extract (IOE) significantly increased the proliferation of myoblast compared to the non-treated cells. However, IOE did not intensify cell proliferation than the cells treated with Octacosanol (OCT, µg/mL) used as a positive control. Four other extracts [*Ecklonia cava* extract (ECE), *Hizikia fusimorme* extract (HFE), *Myelophcus caespitosus* extract (MCE), and *Sargassum horneri* extract (SHE)] did not affect cell proliferation on C2C12 cells (Figure 4).

### 2.3. In Silico Docking Simulation of Marine-Derived Bioactive Compounds to TNF-α

Bioactive compounds possess multi-functional activities based on their structural characteristics such as hydrophobicity, charge, microelement binding activity. To explore a TNF-α inhibitor from marine-derived products, the biological network dynamic between TNF-α and these marine-derived natural compounds were simulated in computational space. The crystal structure of TNF-α was allocated from Protein Data Bank (PDB, http://www.pdb.org) (PDB ID: 2AZ5). Among the five marine algae extracts, we selected IOE and isolated bioactive compounds, Diphlorethohydroxycarmalol (DPHC) and Ishophloroglucin A (IPA) from IOE, which exhibit both anti-inflammatory and skeletal muscle cell proliferation activities which might be TNF-α inhibitor candidates materials. They are polyphenols of phlorotannin types, which are repeated units of phloroglucinol. Their two-dimensional structures are indicated in Figure 5. As shown in Figure 6, of the TNF-α inhibitor candidates, only the DPHC was stably bound to TNF-α (2AZ5) in a total of 4 poses. DPHC-2AZ5 was stable combined with the lowest binding energy value (−53.73 kcal/mol) and CDOCKER interaction energy value (−40.33 kcal/mol). However, IPA could not bind TNF-α (2AZ5), and therefore, we could not obtain the binding form, the binding energy and CDOCKER interaction energy value. The OCT used as a positive control was combined with more diverse poses than the number of poses in which DPHC was combined with 2AZ5. However, the lowest binding energy value (−44.01 kcal/mol) and CDOCKER interaction energy value (−31.83 kcal/mol) of the total 49 poses of OCT-2AZ5 was higher than that of DPHC-2AZ5 (Table 1).

### 2.4. Effects of DPHC on NO Production and Pro-Inflammatory Cytokines in TNF-α-Induced Inflammatory Myopathy Cells

NO is a representative indicator, one of various inflammatory factors, that is inflammatory responses against various inflammation stimuli such as pro-inflammatory cytokines and oxidative stresses. Therefore, the protective effect of DPHC was assessed for the production of NO in skeletal muscle cells stimulated by TNF-α. The level of NO generation is significantly increased in the TNF-α-stimulated cells compared to the untreated cells (Figure 7). However, when the NO production produced in C2C12 cells by TNF-α was calculated as 100.00%, NO production was significantly inhibited to 85.03%, 78.91%, 74.15%, and 74.15% due to treatment of each concentration (1.56, 3.125, 6.25, and 12.5 µg/mL) of DPHC. (Figure 7). The inflammation process is activated by inflammatory stimulation secrets NO, prostaglandin, and cytokines such as interleukin-1β (IL-1β), interleukin-6 (IL-6), and TNF-α. These pro-inflammatory mediators play an important role in various inflammatory diseases targets. To evaluate the protective effects of DPHC on TNF-α-induced inflammatory myopathy, the levels of inflammatory cytokines, including IL-1β, IL-6, and TNF-α mRNA were analyzed by qRT-PCR. As a result, the expression of levels of TNF-α, IL-1β, and IL-6 mRNA increased to about 2.40, 2.28, and 2.43-folds in TNF-α-treated cells, respectively, compared to the control. However, their mRNA expression levels were reduced to about 0.41, 0.68, and 0.57-folds by DPHC pre-treatment (at 12.5 µg/mL). The results clearly demonstrated that the expression of pro-inflammatory cytokines mRNA was sufficiently suppressed by DPHC treatment compared to the TNF-α-stimulated cells (Figure 7).

### 2.5. Protective Effects of DPHC on Inflammatory Myopathy through Nuclear Factor-κB (NF-κB) and Mitogen-Activated Protein Kinase (MAPKs) Signaling Pathways

To investigate whether the regulation of inflammatory response by DPHC mediated through a NF-κB and MAPK pathways, phosphorylation of NF-κB (P-NF-κB) and MAPK protein expressions were analyzed. As shown in Figure 8, TNF-α significantly promoted the protein expression levels of p-inhibitor of nuclear factor kappa B (IκB)α and p-p65 NF-κB to 1.58 and 1.53-folds, respectively compared to the non-treated in C2C12 cells. However, due to DPHC pre-treatment, the protein expression level of p-IκBα decreased to 0.62-fold and 0.42-fold (at 6.25, and 12.5 µg/mL, respectively). In addition, the expression level of p-p65 NF-κB protein was decreased to 0.51-fold at a concentration of 6.25 µg/mL DPHC, and p-p65 NF-κB protein was not expressed at a high concentration of DPHC (12.5 µg/mL). Especially, DPHC suppressed the p-p65 NF-κB expressions increased by TNF-α. The phosphorylations of MAPKs [c-Jun N-terminal kinase (JNK) and p38)] were not stimulated by TNF-α, but DPHC suppressed the p-JNK and p-p38. These results suggest that TNF-α induces inflammatory myopathy through p65 NF-κB rather than MAPKs and DPHC also represents protective effects via p65 NF-κB pathways. In addition, to determine whether DPHC can inhibit TNF-α-induced the Muscle RING-finger protein (MuRF)-1 and Muscle Atrophy F-box (MAFbx)/Atrogin-1 expression in myocytes, MuRF-1 and MAFbx/Atrogin-1 protein expressions were analyzed. As shown in Figure 8, TNF-α significantly promoted the protein expression levels of MuRF-1 and MAFbx/Atrogin-1 to 1.40-fold and 1.15-fold, respectively, compared to non-treated cells, but DPHC (12.5 µg/mL) significantly inhibited the protein expression levels of the MuRF-1 and MAFbx/Atrogin-1 promoted by TNF-α to 0.60-fold and 0.56-fold. These results indicate that TNF-α clearly causes inflammatory myopathy through NF-κB rather than MAPKs pathways and DPHC has protective effects against inflammatory myopathy.

## 3. Discussion

Inflammation is an important host defense system response to external physical or chemical infection and injury, and it plays a role in maintaining health state against variety stimuli. The inflammatory process is regulated by inflammatory cells including macrophages, neutrophils, eosinophils, and mononuclear phagocytes [18]. Normal inflammatory responses are mediated by a correlation between down-regulation of pro-inflammatory proteins and up-regulation of anti-inflammatory proteins [19]. Inflammation can be classified into two phases as acute and chronic. The acute phase is associated with the accumulation of fluids, elevated blood flow, the increase of the number of leukocytes and inflammatory mediators, whereas the chronic inflammation is associated with progression of specific humeral and cellular immune responses [20]. Inflammation affects various organs of the body, and also is associated with disruption of anabolic signals initiating muscle growth. Therefore, inflammation is an important contributor to the pathology diseases implicated in skeletal muscle dysfunction [21]. While acute or chronic inflammation disease states exhibit different pathologies, all have in common the loss skeletal muscle mass and a deregulated skeletal muscle physiology. In the acute inflammation, neutrophils and pro-inflammatory macrophages (M1) massively accumulated in the injured muscle and the M1 macrophages release a variety of pro-inflammatory agents including TNF-α [22,23]. In chronic inflammation, the persistence of neutrophils impairs macrophage conversion from M1 to M2 macrophages profile and these cells adopt a hybrid phenotype, which impairs muscle healing and triggers fibrosis by releasing an exaggerate amount transforming growth factor (TGF)-beta [23,24,25,26]. TNF-α is an inflammatory cytokine implicated in muscle atrophy conditions associated with various diseases [26]. TNF-α belongs to a family of secreted cell surface proteins that mediate immune and inflammatory responses [27]. Anti-TNF-α biologics currently on the market for the treatment of inflammatory disease [28]. The synthetic antibodies such as etanercept, infliximab, and adalimumab, approved for the treatment of inflammatory diseases bind to TNF-α directly, and leads to prevent its association with the TNF receptor [29]. However, despite considerable developments, they have the potential to cause serious side effects such as eliciting an autoimmune anti-antibody response or the weakening of the body’s immune defenses to opportunistic infections [29]. Therefore, it is important to be able to regulate the excessive released TNF-α to prevent it from working in skeletal muscle. Thus, we focused on finding molecules, marine brown algae-derived biological substances that could directly inhibit TNF-α.

Prior to carrying out the potential ability to act as a TNF-α inhibitor in the natural biological substances from marine brown algae, we examined both the anti-inflammatory activity and skeletal muscle cell proliferation activity of brown algae extracts in vitro. In the present study, five species of marine brown algae collected from Jeju Island were used for screening experiments. All separated from algal extracts showed non-cytotoxic effect on RAW 264.7 macrophages and C2C12 skeletal myocytes. In addition, they dramatically inhibited NO production against LPS-induced inflammation. Especially, the IOE significantly induced skeletal muscle cell proliferation compared to non-treated cells. However, ECE, HFE, MCE, and SHE did not affect cell proliferation on C2C12 cells. From these results, therefore, we explored biological substances within *I*. *okamurae* that indicate anti-inflammatory effects and muscle cell proliferation activity.

*I*. *okamuare*, an edible brown alga called ‘pae’ in South Korea, is widely inhabited from South Korea to China and Japan. In this study, *I*. *okamuare* in the temperate coastal area of Jeju Island of South Korea was used. The *I*. *okamuare* has shown various biological activities in vitro and in vivo, such as antioxidant, antidiabetes, and neuroprotective effects [30,31,32]. Especially, DPHC found in IOE has been consistently reported to represent a variety of biological effectiveness [33,34,35,36]. In particular, DPHC is phlorotannin isolated from the EtOAc fraction of IOE, and DPHC was confirmed to be the main compound from the result of high performance liquid chromatography (HPLC) of the EtOAc fraction [37], and it was reported that it is the main compound showing various bioactivities via HPLC [34,35,36,38]. In addition, DPHC has been reported that exhibit anti-inflammatory activity [39,40,41], and it has also been reported to exhibit various bioactivities by regulating NF-κB signaling pathways, are a major pleiotropic transcription factor modulating immune, inflammatory, cell survival, and proliferative responses [35,42,43,44]. IPA, another novel main phlorotannin identified from *I*. *okamurae*, was recently found to be involved metabolism such as diabetes and fat accumulation [45,46,47]. Thus, we selected DPHC, which exhibits anti-inflammatory activity and regulates NF-κB signaling pathways, as the candidate material. The IPA identified in the *I*. *okamurae* has not been reported to be involved in inflammation relief, but it has been selected as a candidate because it has recently been newly identified as another main phlorotannin in *I*. *okamurae*. Numerous reports have associated TNF-α and NF-κB with muscle wasting diseases and myopathy [48,49,50]. In addition, Bakkar et al. [51] reported that binding of TNF-α to its receptor, initiates a IKK-γ dependent signaling cascade that activates the inactive p50/p65 heterodimer and causes its translocation into the nucleus where it decreases the expression of the pro-myogenic transcription factor, MyoD. Therefore, in order to prevent the combination of TNF-α and its receptor, we performed in silico molecular docking simulation tool to determine whether the candidates DPHC and IPA can be bind into their binding site. Molecular docking has been used in several studies [52,53,54] as an efficient computational method to study interaction patterns or predict potential binding modes of small molecules or ligands within the active site of a known three-dimensional protein or receptor for drug design [55]. In particular, it may be used to predict the strength of association or binding affinity between two molecules using binding and interaction energies [53]. In this study, we obtained a molecular model from PDB that could explore TNF-α inhibitors and used it as a model [56,57]. As a results, especially, TNF-α-ligands complexes were well-performed with DPHC and OCT (positive control) stably posed in the pocket of the TNF-α. The binding energy and CDOCKER interaction energy value respectively represent the following meaning. The electron is the energy value when there is no shape change (angle, etc.) in the structure of the receptor, and only the structure of the ligand is changed. The latter is energy value when changes occur in both the structure of the receptor and ligand. Because their energy values are expressed in negative form, the lower the energy values, the stronger the coupling is. The docking analysis results indicated that the DPHC stably binds to TNF-α with lowest binding energy value (−53.73 kcal/mol) and CDOCKER interaction energy (−40.33 kcal/mol) compared to OCT, positive control (binding energy, −44.01 kcal/mol; CDOCKER interaction energy, −31.83 kcal/mol). However, IPA has a result that does not combine with TNF-α.

As mentioned above, recent studies have identified NF-κB as an important transcription factor involved in skeletal muscle atrophy caused by TNF-α [48,49,50]. In differentiating myoblasts, TNF-α induced activation of NF-κB led the reduced expression of the differentiation markers myogenin and myosin chain [48]. In addition, Li [58] have demonstrated that an increase the expression of MuRF-1 and MAFbx/Atrogin-1 proteins through NF-κB and MAPKs pathways in C2C12 cells exposed to TNF-α. NF-κB also inhibits satellite cell activation and differentiation by inhibiting MyoD expression [49]. Overexpression of MAFbx/Atrogin-1 in cultured C2C12 myotubes produces cell atrophy and mice knocked out of the MAFbx/Atrogin-1 gene is more resistant to muscle wasting that results from muscular denervation than their wild-type littermates [59]. This evidence suggests that MAFbx/Atrogin-1 plays a pivotal role in TNF-α-induced muscle atrophy. In this study, therefore, we analyzed the protective activity of DPHC against TNF-α-induced inflammatory myopathy through p65 NF-κB and MAPKs pathways in C2C12 cells. Prior to this work, we isolated DPHC from IOE. However, the method used by previous researchers to isolated and purify DPHC requires various and complex processes, such as the use of silica gel, Sephadex-LH20 column chromatography, and preparative HPLC [34,35,36,38]. These conventional methods have several disadvantages such as takes a long time, requires a limited amount of compound, as well as target compounds, are irreversibly adsorbed on the stationary phase during separation [9]. In this study, therefore, we isolated DPHC from IOE using high performance centrifugal partition chromatography (HPCPC) to compensate for these shortcomings. HPCPC is one of the best techniques for purification with a higher sample loading capacity, accuracy, and speed; less solvent use; and reproducible separation of target compounds in a single step process [37]. DPHC was isolated from IOE using HPCPC as follows according to the method described in Kim et al. [37] and evaluated DPHC activity against TNF-α-induced inflammatory myopathy. As the results, DPHC down-regulated the protein expression levels of MuRF-1 and MAFbx/Atrogin-1, which are the key protein of muscle atrophy through NF-κB and MAPKs signaling pathways in TNF-α-stimulated inflammatory myopathy C2C12 cells. Therefore, it was expected that DPHC exhibit better TNF-α inhibitory activity and lead to protect the skeletal muscle degradation by TNF-α.

## 4. Materials and Methods

### 4.1. Chemicals and Reagents

Dulbecco’s Modified Eagle’s Medium (DMEM), fetal bovine serum (FBS), horse serum (HS), penicillin-streptomycin, trypsin-EDTA, and Dulbecco’s Phosphate Buffered Saline (DPBS) were purchased from Gibco-BRL (Burlington, ON, Canada) All the other chemicals used were of analytical grade.

### 4.2. Preparation of Crude Extract and Bioactive Compound from Marine Brown Algae

Five species of marine brown algae were collected on the coast of Jeju Island (Table 2). The dried algae (500 g) were extracted with 80% (*v*/*v*) aqueous methanol (5 L) under stirring at room temperature (RT) for 24 h. The extract was evaporated in vacuo. The yield of the extracts is as shown in Table 2. In order to isolate the bioactive compound DPHC from IOE, the process followed the method used in our previous study [37]. DPHC (0.39%, ≥85% purity, Figure 9) from IOE (14.10%) was compared by previously reported ^1^H and ^13^C NMR data [60]. In addition, QTOF-MS was shown [M+H]^+^ ion at *m*/*z* 513.14 in positive mode.

### 4.3. Anti-Inflammatory Activities of Marine Brown Algal Extracts

#### 4.3.1. Cytotoxic Assessment Using MTT Assay

The murine macrophage cell line RAW 264.7 was purchased from the Korean Cell Line Bank (KCLB, Seoul, Korea). RAW 264.7 cells were cultured in DMEM supplemented with 10% heat-inactivated FBS, streptomycin (100 µg/mL), and penicillin (100 unit/mL) at 37 °C under 5% CO_2_ humidified incubator. Exponential phase cells were used throughout the experiments. Then, RAW 264.7 cells (1.5 × 10^4^ cells/mL) plated in 24-well plates were fore 16 h and then treated with LPS (1 µg/mL) plus aliquots of the five marine brown algal methanol extract. The cells were then incubated for an additional 24 h at 37 °C under 5% CO_2_ humidified incubator. After incubation, the cytotoxic assessment was performed using an MTT assay. The formazan crystals were dissolved in dimethyl sulfoxide (DMSO), and the absorbance was measured using an ELISA plate reader at 540 nm (BioTek Instruments, Inc., Winooski, VT, USA). The optical density of the formazan generated in non-treated control cells was considered to represent 100% viability.

#### 4.3.2. Determination of Nitric Oxide (NO) Production

RAW 264.7 cells (1.5 × 10^4^ cells/mL) were seeded in 24-well plates and incubated for 16 h and then treated with five marine brown algae methanol extracts for 1 h and stimulated LPS (1 µg/mL) for 24 h. After, the quantity of nitrite accumulated in the culture medium was measured as an indicator of NO production using the protocol described in [10].

### 4.4. Skeletal Muscle Cell Proliferation Activities of Marine Brown Algae

#### 4.4.1. Myoblast Cell Culture and Differentiation

The murine C2C12 skeletal myoblasts were obtained from American Type Culture Collection (ATCC, Manassas, VA, USA) were cultured in DMEM supplemented with 10% heat-inactivated FBS, streptomycin (100 mg/mL), and penicillin (100 unit/mL) at 37 °C under 5% CO_2_ humidified incubator. To induce differentiation, 80% confluent cultures were switched to DMEM containing 2% HS for 4 days with medium changes every other day.

#### 4.4.2. Cytotoxic Assessment and Cell Proliferation Activity

The cytotoxic assessment was performed using MTT assay. The formazan crystals were dissolved in DMSO, and the absorbance was measured using an ELISA plate reader at 540 nm (BioTek Instruments, Inc., Winooski, VT, USA). The optical density of the formazan generated in non-treated control cells was considered to represent 100% viability. To assess the skeletal muscle cell proliferation effects of five marine brown algae extract, activities were determined using BrdU assay (Millipore, Billerica, MA, USA). C2C12 cells (5.0 × 10^4^ cells/mL) plated in 48-well plates were pre-incubated for 48 h and then converted into a differentiation medium (DMEM containing 2% HS) and treated with five marine brown algae methanol extracts every other day. Cell proliferation was calculated by comparison with the absorbance at 450 nm of standard solutions of BrdU in the non-treated cells.

### 4.5. Protective Effects of Marine Biological Active Compound against TNF-α-Induced Inflammatory Myopathy

#### 4.5.1. In Silico Docking Study of New Inhibitor Candidates to TNF-α

To verify the marine brown algae-derived bioactive compounds can act acts as TNF-α inhibitor, molecular docking studies including the docking simulation of the protein-ligand complex, the possibility of binding, and value of binding energy were performed using flexible docking tool in Accelrys Discovery Studio (DS) 3.5 (BIOVIA, CA, USA). The crystal structure of TNF-α was allocated from PDB and shown in Figure 5. The structural information of candidate compounds was provided from our previous studies [37,45,60], and the structures were drawn by ChemDraw Ultra 12.0 (Chemistry.Com.Pk, Figure 5). For molecular docking analysis, we performed the following steps: (1) conversion of the 2D structure into 3D structure of ligands; (2) preparing protein and defining the binding site; (3) preparing ligands; (4) flexible docking simulation; (5) calculation binding energies.

#### 4.5.2. Determination of NO Production and Pro-Inflammatory Cytokines Expression against TNF-α-Induced Inflammatory Myopathy Cells

To determine the inhibitory effects of natural biological active substance from IO on NO production and pro-inflammatory cytokines expression in inflammatory C2C12 myotubes, C2C12 cells were differentiated with new inhibitor candidate and then induced inflammation using TNF-α. NO production was detected using Griess assay and pro-inflammatory cytokines (TNF-α, IL-1β, and IL-6) total RNA expression was analyzed using quantitative RT-PCR (qRT-PCR, Roche, Basel, Switzerland).

#### 4.5.3. Western Blot Analysis of New Inhibitor Candidate against TNF-α-Induced Inflammatory Myopathy Cells

In order to verify the effects of natural biological active substance from IO as new inhibitor of TNF-α, protein expression levels in the cytoplasm and nucleus, which are key molecules involved in skeletal muscle atrophy were analyzed. Nucleic and cytoplasmic proteins were extracted from the cells, and p-IκB-α, p-p65NF-κB, MAPKs (p-JNK, and p-p38), MuRF-1, and MAFbx/Atrogin-1 protein expression were detected using specific primary rabbit/mouse polyclonal antibodies and goat anti-rabbit or –mouse IgG HRP conjugated secondary antibodies. Signals were developed using an enhanced chemiluminescence (ECL) Western blotting detection kit (Amersham, Arlington Heights, IL, USA) and exposed to FUSION Solo6S program equipped with eVo-6 camera (Vilber Lourmat, Marne-La-Vallée, France). The basal levels of each protein were normalized by analyzing the level of β-Actin or Lamin B protein by using the ImageJ program.

### 4.6. Statistical Analysis

All experiments were conducted in triplicate (*n* = 3) and an one-way analysis of variance (ANOVA) test (using SPSS 12.0 statistical software) was to analyze the data. Significant differences between the means of parameters were determined by using Tukey test to analyze the difference. *p*-values of less than 0.05 (*p* < 0.05) and 0.01 (*p* < 0.01) was considered as significant.

## 5. Conclusions

DPHC was identified as a TNF-α inhibitor agent in a structure-based virtual analysis study. In addition, The IOE that contains the DPHC indicated that the anti-inflammatory effects and skeletal muscle cell proliferation activity without cytotoxicity. Furthermore, DPHC suppressed the inflammatory myopathy related protein expression through NF-κB (p-IκB-α/p-p65NF-κB), and MAPKs (p-JNK/p-p38) signaling cascades. It was expected that DPHC is therefore a promising candidate for the development of safe pharmacological agent that have potent inhibitory effects against inflammatory myopathy by TNF-α instead of synthetic drugs.

## Figures and Tables

**Figure 1 marinedrugs-18-00529-f001:**
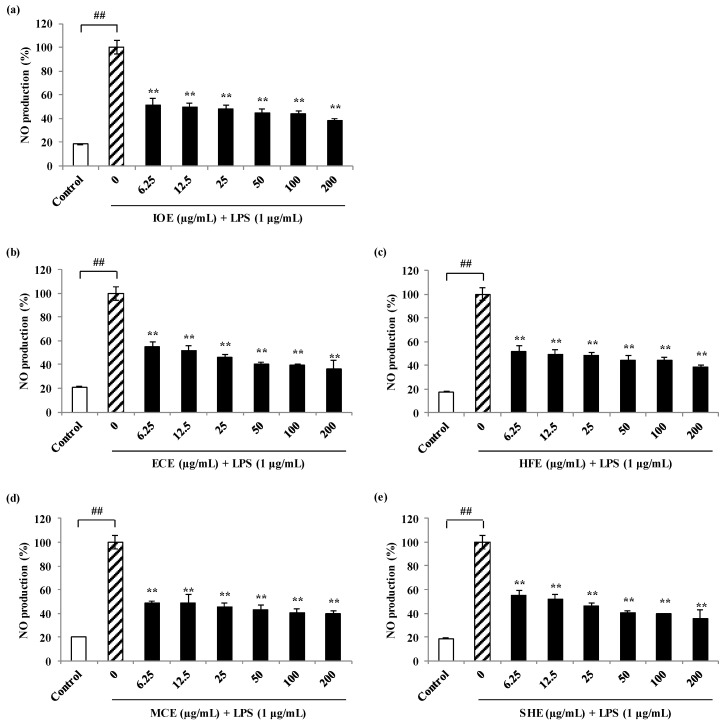
Anti-inflammatory effects of marine brown algal extracts against LPS-stimulated RAW 264.7 macrophage cells. Inhibition effects of IOE (**a**), ECE (**b**), HFE (**c**), MCE (**d**), and SHE (**e**) on NO production against LPS-induced cells. Experiments were performed in triplicate and the data were expressed as mean ± S.E.M.; ## *p* < 0.01 as compared to the untreated group. ** *p* < 0.01 as compared to the LPS-treated group. IOE, *Ishige okamurae* extract; ECE, *Ecklonia cava* extract; HFE, *Hizikia fusiforme* extract; MCE, *Myelophcus caespitosus* extract; SHE, *Sargassum horneri* extract.

**Figure 2 marinedrugs-18-00529-f002:**
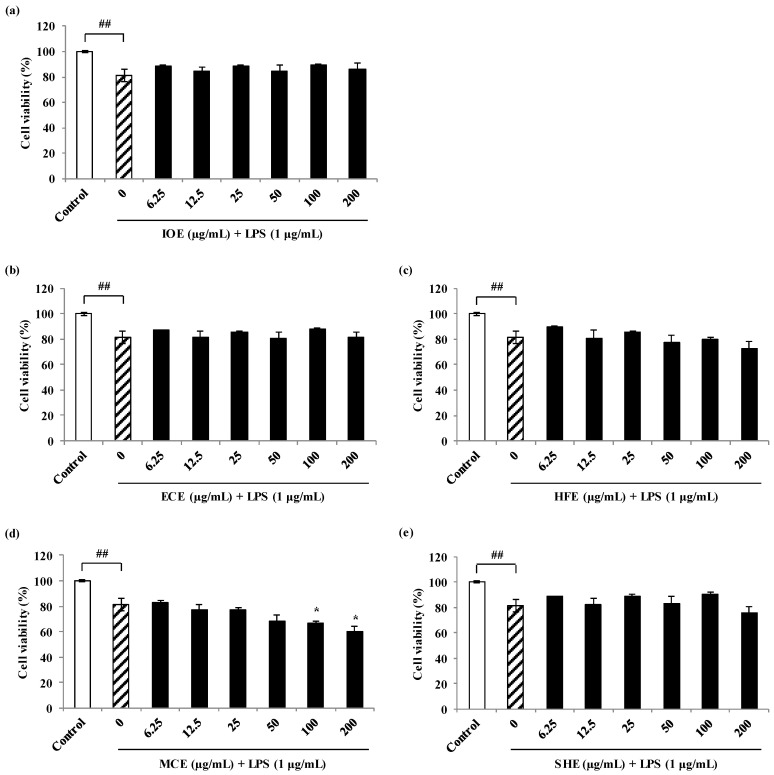
Cell protective effects of marine brown algal extracts against LPS-stimulated inflammatory RAW 264.7 macrophage cells. Cell viability of IOE (**a**), ECE (**b**), HFE (**c**), MCE (**d**), and SHE (**e**) on LPS-induced inflammatory cells. Experiments were performed in triplicate and the data were expressed as mean ± S.E.M.; ## *p* < 0.01 as compared to the untreated group. * *p* < 0.05 as compared to the LPS-treated group. IOE, *Ishige okamurae* extract; ECE, *Ecklonia cava* extract; HFE, *Hizikia fusiforme* extract; MCE, *Myelophcus caespitosus* extract; SHE, *Sargassum horneri* extract.

**Figure 3 marinedrugs-18-00529-f003:**
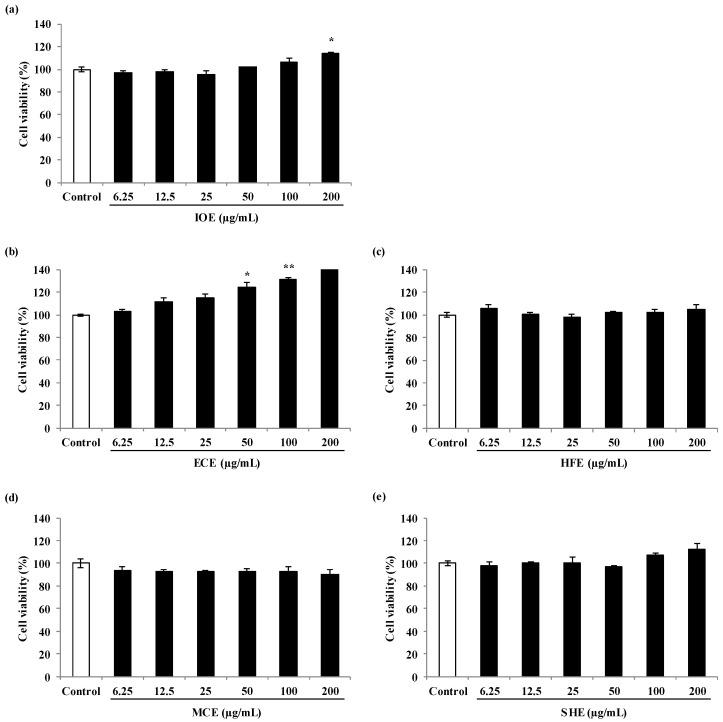
Cell toxicity of marine brown algal extracts on C2C12 skeletal myoblast cells. (**a**), IOE-treated cells; (**b**), ECE-treated cells; (**c**), HFE-treated cells; (**d**), MCE-treated cells; (**e**), SHE-treated cells. Experiments were performed in triplicate and the data were expressed as mean ± S.E.M.; * *p* < 0.05, and ** *p* < 0.01 as compared to the untreated group. IOE, *Ishige okamurae* extract; ECE, *Ecklonia cava* extract; HFE, *Hizikia fusiforme* extract; MCE, *Myelophcus caespitosus* extract; SHE, *Sargassum horneri* extract.

**Figure 4 marinedrugs-18-00529-f004:**
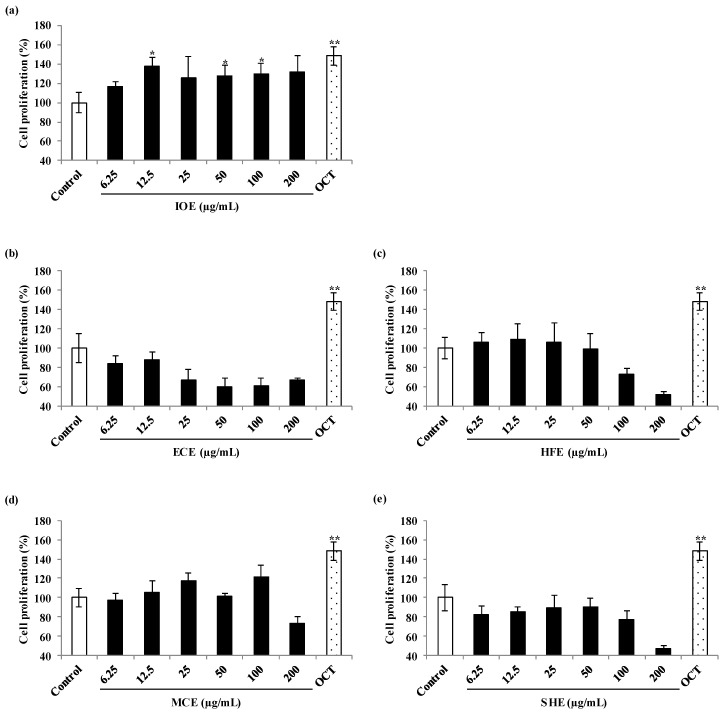
Skeletal cell proliferation activities of marine brown algal extracts on C2C12 skeletal myoblast cells during differentiation periods. (**a**), IOE-treated cells; (**b**), ECE-treated cells; (**c**), HFE-treated cells; (**d**), MCE-treated cells; (**e**), SHE-treated cells. Experiments were performed in triplicate and the data were expressed as mean ± S.E.M.; * *p* < 0.05, and ** *p* < 0.01 as compared to the untreated group. IOE, *Ishige okamurae* extract; ECE, *Ecklonia cava* extract; HFE, *Hizikia fusiforme* extract; MCE, *Myelophcus caespitosus* extract; SHE, *Sargassum horneri* extract; OCT, Octacosanol as a positive control (µg/mL).

**Figure 5 marinedrugs-18-00529-f005:**
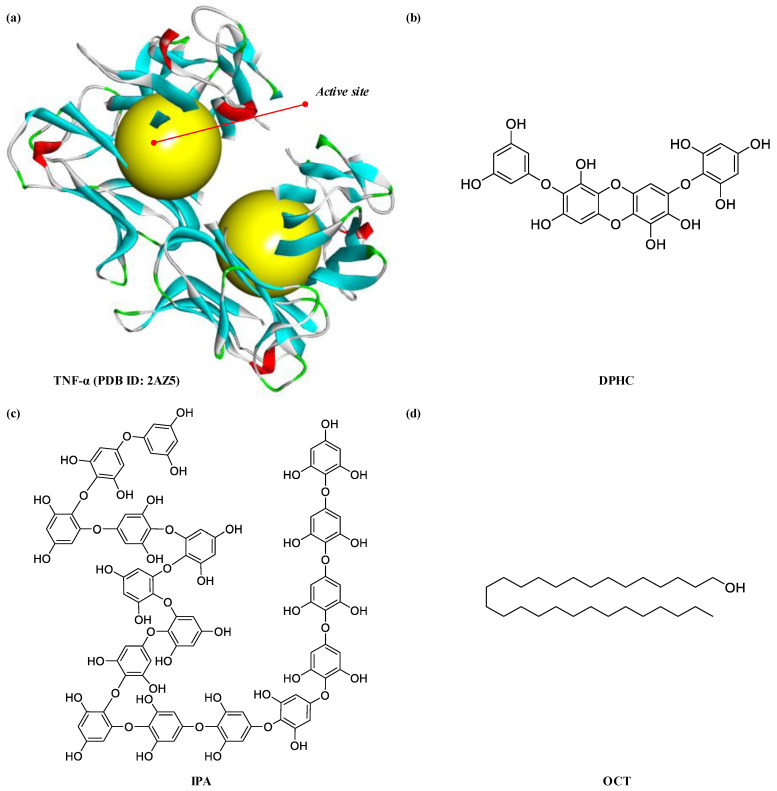
Crystal structure of TNF-α allocated from protein data bank (PDB ID: 2AZ5) (**a**). (**b**–**d**), The structure of marine brown algal-derived compounds and OCT. Structure of DPHC, IPA, and OCT were drawn by ChemDraw Ultra 12.0 (Chemistry.Com.Pk) and obtained by converting to 3D structure in the Accelrys DS 3.5 program. (**b**), Diphlorethohydroxycarmalol (DPHC); (**c**) Ishophloroglucin A (IPA); (**d**), Octacosanol (OCT, positive control).

**Figure 6 marinedrugs-18-00529-f006:**
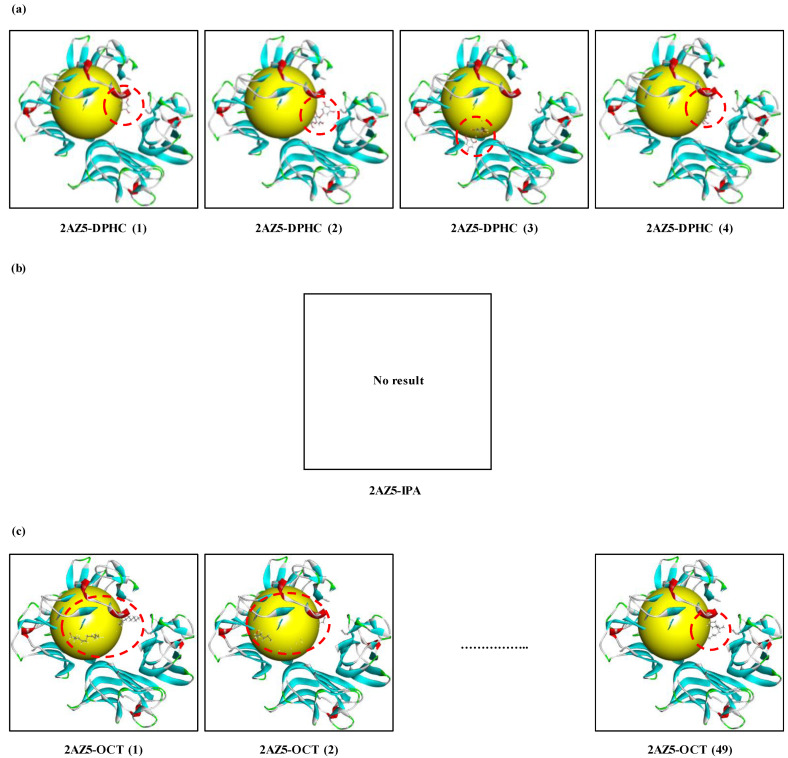
Computational prediction of binding 3-dimensional (3D) structures by docking simulation of 2AZ5, a 3D model of TNF-α provided from Protein Data Bank (PDB), and DPHC, IPA, and OCT (as a positive control). (**a**), 2AZ5-DPHC means a complex of 2AZ5 and DPHC. DPHC was combined with 2AZ5 in four different poses (2AZ5-DPHC (1), 2AZ5-DPHC (2), 2AZ5-DPHC (3), and 2AZ5-DPHC (4)). (**b**), 2AZ5-IPA means a complex of 2AZ5-IPA. However, IPA did not bind to TNF-α. Therefore, we expressed the result picture as no result. (**c**), 2AZ5-OCT means a complex of 2AZ5 and OCT. OCT was combined with TNF-α in a total of 49 poses. We show two complexes combined with the lowest binding energy value (2AZ5-OCT (1), and 2AZ5-OCT (2), respectively), and one complex combined with the highest binding energy value (2AZ5-OCT (3)). Diphlorethohydroxycarmalol (DPHC); Ishophloroglucin A (IPA); Octacosanol (OCT, positive control).

**Figure 7 marinedrugs-18-00529-f007:**
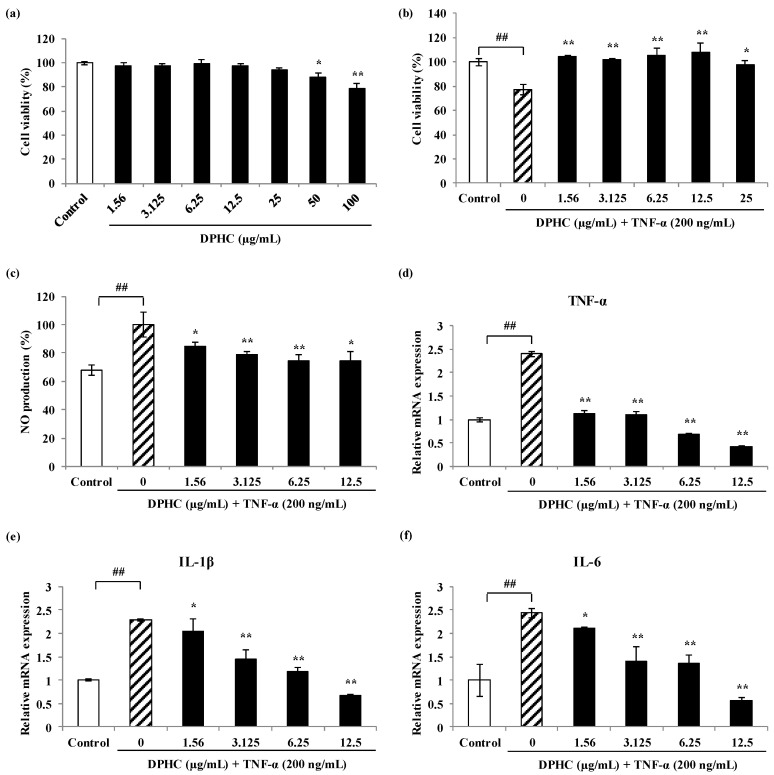
Protective effect of DPHC on TNF-α-induced inflammatory myopathy C2C12 cells. (**a**), Cell toxicity of DPHC on C2C12 myotubes; (**b**), Cell viability of DPHC on TNF-α-induced inflammatory myopathy C2C12 cells; (**c**), NO production by DPHC on TNF-α-induced inflammatory myopathy C2C12 cells; (**d**–**f**), Pro-inflammatory cytokines expression by DPHC on TNF-α-induced inflammatory myopathy C2C12 cells. (**d**), TNF-α cytokine mRNA expression; (**e**), IL-1β cytokine mRNA expression; (**f**), IL-6 cytokine mRNA expression. Experiments were performed in triplicate and the data were expressed as mean ± S.E.M.; ## *p* < 0.01 as compared to the untreated group. * *p* < 0.05, and ** *p* < 0.01 as compared to the TNF-α-treated group.

**Figure 8 marinedrugs-18-00529-f008:**
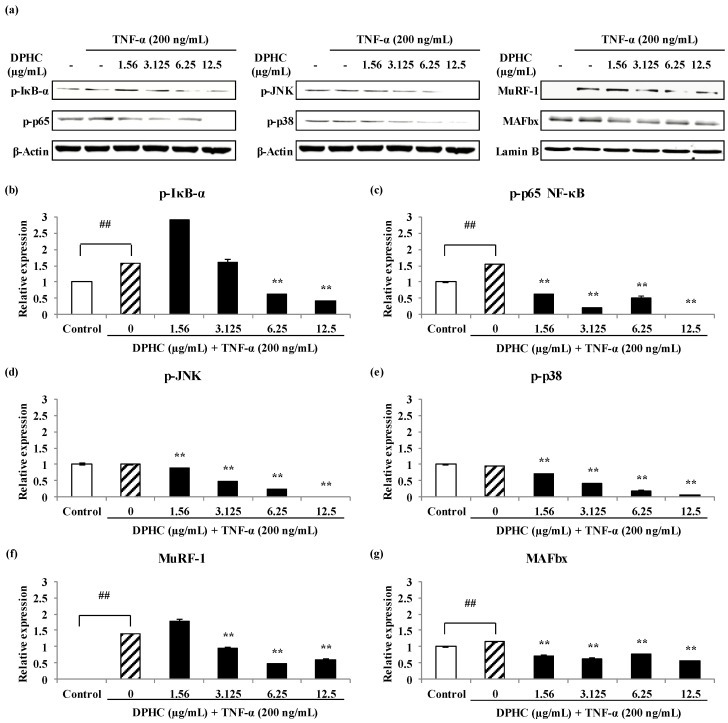
The protein expressions related with inflammatory myopathy in DPHC-treated myotubes against TNF-α-stimulated inflammatory myopathy C2C12 cells. (**a**), Bands of protein expressions; (**b**), p-IκB-α protein expression; (**c**), p-p65 NF-κB protein expression; (**d**), p-JNK protein expression; (**e**), p-p38 protein expression; (**f**), MuRF-1 protein expression; (**g**), MAFbx/Atrogin-1 protein expression. All expression levels of proteins of cytosol (p-IκB-α, p-p65 NF-κB, p-JNK, and p-p38) were measured for β-Actin and those (MuRF-1, and MAFbx/Atrogin-1) in the nucleus for Lamin B by densitometry. Experiments were performed in triplicate and the data were expressed as mean ± S.E.M.; ## *p* < 0.01 as compared to the untreated group. ** *p* < 0.01 as compared to the TNF-α-treated group.

**Figure 9 marinedrugs-18-00529-f009:**
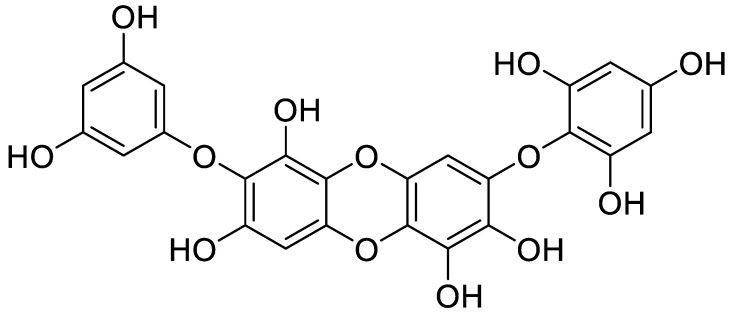
Chemical structure of Diphlorethohydroxycarmolol (DPHC).

**Table 1 marinedrugs-18-00529-t001:** Results of docking simulations of IOE-derived polyphenol compounds with TNF-α (PDB ID: 2AZ5). OCT is octacosanol as a positive control.

Ligands	Total Binding Pose with 2AZ5 (Number)	The Lowest Binding Energy (kcal/mol)	The Lowest-CDOCKER Interaction Energy (kcal/mol)
DPHC	4	−53.73	40.33
IPA	0	0	0
OCT	49	−44.01	31.83

**Table 2 marinedrugs-18-00529-t002:** The list of marine brown algae used in the experiments and the yield of these brown algae extracts.

No.	Scientific Name	Abbreviation of Extract	Yield (%)
1	*Ishige okamurae*	IOE	14.10 ± 0.14
2	*Ecklonia cava*	ECE	20.87 ± 1.89
3	*Hizikia fusiforme*	HFE	7.00 ± 1.83
4	*Myelophcus caespitosus*	MCE	6.30 ± 0.14
5	*Sargassum horneri*	SHE	5.47 ± 0.12

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
