# Peer review of "Diphlorethohydroxycarmalol (DPHC) Isolated from the Brown Alga Ishige okamurae Acts on Inflammatory Myopathy as an Inhibitory Agent of TNF-α"

_marinedrugs, 2020, doi:10.3390/md18110529_

Round 1

Reviewer 1 Report

Interesting study, but I think the manuscript has serious shortcomings.

The title does not fully reflect the contents of the paper. No direct experiments on the binding of DHPC with TNF-alpha were described in the manuscript. Therefore authors should either make the direct experiment on the influence of DPHC on binding TNF-alpha with TNF-R or re-write the manuscript to avoid this claim and make it a hypophysis with some experimental background. Molecular docking is not proving the interaction. This compound was described before by authors to be active in a variety of biological models where TNF -alpha is not involved. Previously they characterized  DPHC as potent α-glucosidase and α-amylase inhibitor, antioxidant, AMPK activator, a positive modulator of expression of Cyclooxygenase-1 and -2, downregulator of the NF-κB Signaling Pathway (without signaling from TNF-alpha).  Downregulation of NF-κB Signaling Pathway could undoubtedly cause all the effects described in the manuscript while the binding of DPHC with TNF-alpha could be a mistake of the modeling.

Additionally, the link between brown algae extracts and DPHC was not properly disclosed by authors. Did they isolate this compound as an active substance in this test? Or they took the only well-known compound that they had in hands?  This should be clarified in the text. Now it looks like two different studies - extracts and DPHC.

Reviewer 2 Report

The manuscript by Seo-Young Kim et al. is focused in isolation of Diphlorethohydroxycarmalol from the Brown Alga Ishige okamurae and study of its inhibition activity against TNF-α. After close evaluation of paper I would recommend revision according to the next points:
1. In Introduction: The introduction is too short. I would suggest to update the phrase "Especially, marine brown algae contain a variety of bioactive compounds including phlorotannins, polysaccharide, and pigments and exhibit different biological activities and potential health benefits such as antioxidant activity, anti-inflammatory activity, anti-cancer activity [9], anti-diabetic activity [10], anti-hypertensive effects [11], and anti-obesity activity [12]" with recently published articles: https://www.mdpi.com/1660-3397/18/5/275; https://www.mdpi.com/1660-3397/17/8/434.

2. In Introduction: authors have isolated the Diphlorethohydroxycarmalol. there are not too meny articled describing isolation of phlorotannins. I would suggest to include a special paragraph in Introduction with description of different methods for phlorotannins isolation including , but not limited with these papers: https://doi.org/10.3390/md16120503; https://doi.org/10.1007/s10811-019-02013-2; https://doi.org/10.1007/s11094-019-01987-0.

3. In Section 2: the phrase "All the ethanol extracts 61 showed no cytotoxic effect on RAW 264.7 cells at the tested concentrations (data not shown)" need clarification. Please indicate concentrations range

4. The legend for Fig.1 need clarification of abbreviations "...effects of IO (a), EC (b), HF (c), MC (d), and SH". These abbreviations were not used previously.

5. Please clarify abbreviation OCT (see legend Fig.4).

6. In Scetion 2.3: the phrase "...Bioactive compounds, Diphlorethohydroxycarmalol (DPHC) and Ishophloroglucin A (IPA) derived from IO that exhibit 115 both anti-inflammatory and skeletal cell proliferation activities...' need clarification. In current study authors have not described isolation of Ishophloroglucin A  from IO, and have not studied its anti-inflammatory activity.

7. The Fig. 6 looks confusing. No picture for 2AZ5-IPA, problem with the buttom line.

8. In Table 1 please indicate what is mind Pose (number)?

9. In Section 2.4: the phrase "DPHC significantly inhibited the TNF-α-induced NO production in C2C12 cells.." please provide onformation for how many % inhibited? at which concentration?

10. In section 2.4 please update phrase "The results clearly demonstrated that the expression of proinflammatory cytokines mRNA was sufficiently suppressed by DPHC treatment" with data in numbers, for how many % was suppressed,? at which concentration?

11. In Section 2.4 please support the phrase "As shown in Figure 8, TNF-α significantly promoted the p-IκBα and p-p65 NF-κB in C2C12 cells." with numbers.

12. In secton 2.4. Please support the phrase "However, DPHC decreased the p-IκBα and p-p65 NF-κB expressions." with real data in numbers.

13. Please support the phrase "As shown in Figure 8, TNF-α significantly promoted the MuRF-1 and MAFbx/Atrogin-1 expression, but DPHC significantly inhibited the expression of the MuRF-1 and MAFbx/Atrogin-1." with number, how significantly? at which concentrations?

14. In Discussion: It was shown previously that fucoidans have potent anti-inflammatory activity (https://doi.org/10.4162/nrp.2017.11.1.3; https://doi.org/10.1016/j.ijbiomac.2020.04.012; https://www.mdpi.com/1660-3397/18/5/275). It would be of interest to compare the potential of phlorotannin Diphlorethohydroxycarmalol  with fucoidans.

15. In Section 4.2 please describe, how Diphlorethohydroxycarmalol (DPHC)  was isolated.

16. In Section 4.2: the phrase "...the process followed the method used in our
previous study [44] and compared with data in the literature..." require literature citation. Please compare your method with "data in the literature". 

17. Section 4.2: Please provide information, how compound was identified?

18. In Scetion 4.2 please provide the yield of crude extracts (Table 2) and the yield of DPHC.

19. For all figures please clarify data are presenbted as mean +/- SEM or mean +/- SD?

Reviewer 3 Report

    1. You should check for error in abstract; Ishike okamurae
    2. You should describe full name; TNFa, NF-kB, MAPK etc.
    3. It is recommended to added the information of marine brown algae in Introduction or dicussion; academic name, family, constituents etc.
    4. I think that value (4.0) of Y axis in Figure 8 is too high and suggest to lower as the value (3.0).

Round 2

Reviewer 1 Report

I regret to say that the authors decided to ignore my comments.

I did not find any proofs of direct interaction of TNF-alpha and DPHC but I still see this claim in the title.

As I wrote in the first review any of the effects could be explained by influence on a variety of cellular targets. The molecular docking could not be proof of direct interaction. 

I think the manuscript is not suitable for publication in the present form.

Authors should re-write the manuscript to avoid the claim that DPHC binds to TNF-alpha as they refused to perform direct experiments. 

Reviewer 2 Report

Authors have revised the paper, but some my suggestions were nt addressed correctly.

  1. In Introduction: authors have isolated the Diphlorethohydroxycarmalol. there are not too many articled describing isolation of phlorotannins. I would suggest to include a special paragraph in Introduction with description of different methods for phlorotannins isolation including , but not limited with these papers: https://doi.org/10.3390/md16120503; https://doi.org/10.1007/s10811-019-02013-2; https://doi.org/10.1007/s11094-019-01987-0.
  2. The Fig. 6 is still confusing. Furst, the legend is not self-informative (abbreviations are not clarified). Second: the part 9b0 could be excluded and commented n the text or legend. The numbers in brackets after 2AZ5-DPHC or 2AZ5-OCT not clarified. The number of dots in the figure looks not clear.
  3. In phrase "As a result, the expression of levels of TNF-α, IL-1β, and IL-6 mRNA increased to about 2.40, 2.28, and 2.43...' please indicate units for numbers.
  4. In phrase "However, their mRNA expression levels were reduced to about 0.41, 0.68, and 0.57 levels...' which units you mind under 'level'.
  5. In phrase "DPHC reduced the expression levels of MuRF-1 and MAFbx/Atrogin-1 proteins promoted by TNF-α stimulation to 0.60 level and 0.56 level" what is mind 'level'?
  6. In Section 4.2 please describe shortly, how DPHC was isolated.
  7. Please unify in the text, figures and tables the use of abbreviations: IO or IOE; EC or ECE, HF or HFE, MC or MCE, Sh or SHE

Round 3

Reviewer 1 Report

Dear Authors

I think Marine Drugs can give you plenty of time to prove the fact that DPHC is the TNF-alpha inhibitor. Please contact the Editor.  And I regret to say again that molecular docking is not enough for such a claim.

 My first review still actual  -  No direct experiments on the binding of DHPC with TNF-alpha were described in the manuscript. Therefore authors should either make the direct experiment on the influence of DPHC on binding TNF-alpha with TNF-R or re-write the manuscript to avoid this claim and make it a hypophysis with some experimental background.

Author Response

The main point is that "I regret to say again that molecular docking is not enough for such a claim. My first review still actual  -  No direct experiments on the binding of DHPC with TNF-alpha were described in the manuscript."
The effect of DHPC as a TNF-alpha inhibitor could be proven by molecular docking and in vitro direct binding interaction or indirect experiment. 
TNF-alpha is not an enzyme (enzymes are easily possible to show direct interaction effect), so we could not try the direct binding interaction.
However we proved the inhibition effect of DHPC by in direct experiments by which we produced all the results in the manuscript.
We do say it is absolutely no problem. I am sure this manuscript is enough to prove the title and the aim.
I hope you and the reviewer understand it.

Thank you in advance.
Warm regards,

Jeon

Reviewer 2 Report

Authors have revised the paper according to most of my comments.

Author Response

Thank you.

And, We attach the proof of English correction of this manuscript.

This manuscript is a resubmission of an earlier submission. The following is a list of the peer review reports and author responses from that submission.